# BRD3 Regulates the Inflammatory and Stress Response in Rheumatoid Arthritis Synovial Fibroblasts

**DOI:** 10.3390/biomedicines11123188

**Published:** 2023-11-30

**Authors:** Tanja Seifritz, Matthias Brunner, Eva Camarillo Retamosa, Malgorzata Maciukiewicz, Monika Krošel, Larissa Moser, Thomas Züllig, Matija Tomšič, Oliver Distler, Caroline Ospelt, Kerstin Klein

**Affiliations:** 1Center of Experimental Rheumatology, Department of Rheumatology, University Hospital Zurich, University of Zurich, 8091 Zurich, Switzerlandeva.camarilloretamosa@usz.ch (E.C.R.);; 2Department of Rheumatology and Immunology, Inselspital, Bern University Hospital, University of Bern, 3008 Bern, Switzerland; 3Department for BioMedical Research, University of Bern, 3008 Bern, Switzerland; 4Department of Rheumatology, University Medical Centre Ljubljana, 1000 Ljubljana, Slovenia; 5Faculty of Medicine, University of Ljubljana, 1000 Ljubljana, Slovenia

**Keywords:** synovial fibroblast, arthritis, bromo domain, inflammation, stress response

## Abstract

Background: Individual functions of members of the bromodomain (BRD) and extra-terminal (BET) protein family underlying the anti-inflammatory effects of BET inhibitors in rheumatoid arthritis (RA) are incompletely understood. Here, we aimed to analyze the regulatory functions of BRD3, an understudied member of the BET protein family, in RA synovial fibroblasts (FLS). Methods: BRD3 was silenced in FLS prior to stimulation with TNF. Alternatively, FLS were treated with I-BET. Transcriptomes were analyzed by RNA sequencing (RNAseq), followed by pathway enrichment analysis. We confirmed results for selective target genes by real-time PCR, ELISA, and Western blotting. Results: BRD3 regulates the expression of several cytokines and chemokines in FLS, and positively correlates with inflammatory scores in the RA synovium. In addition, RNAseq pointed to a profound role of BRD3 in regulating FLS proliferation, metabolic adaption, and response to stress, including oxidative stress, and autophagy. Conclusions: BRD3 acts as an upstream regulatory factor that integrates the response to inflammatory stimuli and stress conditions in FLS and executes many functions of BET proteins that have previously been identified using pan-BET inhibitors.

## 1. Introduction

Rheumatoid arthritis (RA) is a chronic, inflammatory joint disease causing cartilage and bone damage that leads to disability and loss of function if not treated well [1]. The most prominent joint-resident cell types are synovial fibroblasts (FLS), which form the lining and sub-lining layers of the synovial membrane in RA [2]. In healthy individuals, the main function of FLS is the production of extracellular matrix and the lubrication of the joint [3]. In RA, FLS are activated and overtake pro-inflammatory and pro-destructive functions [2]. Another characteristic of the RA synovium is the activation of stress response pathways, including oxidative stress and hypoxic stress response pathways, unfolded protein response (UPR), and autophagy [4,5,6,7]. This is accompanied by a metabolic switch in FLS towards glycolysis [8,9] that potentially contributes to the switch from an acute to chronic disease [10]. Together, these pathways enable FLS to survive in and adjust to the altered joint microenvironment in RA, to provide sufficient energy and cope with an accumulation of misfolded proteins concomitant with the production of large amounts of pro-inflammatory cytokines, chemokines, and matrix-degrading enzymes.

Upstream factors that regulate the interplay of inflammatory and stress pathways are potential new targets for the treatment of RA and other chronic inflammatory diseases. Candidates for therapeutic targeting are members of the bromodomain (BRD) protein family, in particular, bromo- and extra-terminal (BET) proteins. The BET protein family consists of four members of double BRD proteins: the ubiquitously expressed members BRD2, BRD3, and BRD4, and the testis-specific BRDT [11]. They are readers of ε-N-acetylation modifications present on histone side chains in promoters and enhancers [12]. BET protein-mediated recognition of activating histone marks and the recruitment of protein complexes to DNA regulatory elements of their target genes are key events in transcriptional regulation [11].

The first pan-BET protein inhibitors, I-BET and JQ-1, were described in 2010 and were shown to have anti-inflammatory and anti-cancer activities [13,14]. Since then, BET inhibitors have been discussed and tested in first clinical trials as potential therapies in numerous diseases, including different types of cancer, type 2 diabetes mellitus, and cardiovascular and coronary artery disease [15].

Anti-inflammatory effects have been extensively studied in different pre-clinical models of arthritis in vivo, and in a range of disease-relevant cell types in vitro [15]. Whereas numerous studies have addressed the effects of BET inhibitors, individual functions of BET proteins have hardly been investigated. Although BET inhibitors are pan-inhibitors, current studies of BET proteins in RA FLS focus on the regulatory roles of BRD2 and BRD4 [16,17] and do not include an analysis of BRD3 functions. In recent years, BET proteins, particularly BRD4, have been shown to regulate stress response pathways, including oxidative stress and heat shock responses, and autophagy [18,19,20,21]. In this study, our aim was to study the role of BRD3 in regulating FLS gene expression. We highlight the functions of BRD3 in regulating cytokine and chemokine production, as well as its role in the metabolic and stress-related adaption of FLS under inflammatory conditions in FLS.

## 2. Materials and Methods

### 2.1. Patient Samples and Cell Preparation

Synovial tissue specimens were obtained from hand, shoulder, and knee joints of RA patients undergoing joint replacement surgery (Schulthess Clinic Zurich, Switzerland). All patients fulfilled the criteria for the classification of RA [22]. FLS were isolated and cultured as described elsewhere [23] and used between passages four and eight for all experiments. The study was approved by the ethics committee of the Canton of Zurich, Switzerland. All patients provided informed consent prior to inclusion in the study.

### 2.2. Silencing of BRD3

FLS were transduced with lentiviral particles targeting BRD3 or control particles (Santa Cruz Biotechnology, Dallas, TX, USA). Forty-eight hours after transduction, cells were stimulated with TNF (10 ng/mL) for 24 h. Knockdown of BRD3 was verified by Western blotting and RNA sequencing (RNAseq).

### 2.3. RNA Sequencing of FLS

Total RNA from FLS (wrist; *n* = 3/group) was isolated using the RNeasy Mini Kit (Qiagen, Hilden, Germany). The RNA quality and quantity were evaluated using the Agilent RNA 6000 Nano Kit and the Agilent 2100 Bioanalyzer instrument (Agilent Technologies, Santa Clara, CA, USA). Library preparation and RNAseq were performed at the Functional Genomics Center Zurich (FGCZ). Libraries for RNAseq were generated using the Illumina TruSeq Stranded Total RNA Sample Preparation Kit. The quality and quantity of the generated libraries were checked using the Agilent 2100 Bioanalyzer instrument and a DNA-specific chip. Libraries were sequenced by Illumina NovaSeq 6000, with single-read approaches (100 bp). RNAseq data were deposited in NCBI’s Gene Expression Omnibus [24] and are accessible through the GEO Series accession number GSE247226.

### 2.4. Analysis of RNAseq Results

Quality control (QC) of RNAseq data was conducted using FASTQC (Wingett, 2018 #1157). Reads were mapped to the human reference genome (hg19) using STAR [25]. Counts per gene were assessed using Feature Counts [26]. We investigated differentially expressed genes between experimental conditions using the “DESeq2” package of Bioconductor. We removed genes with less than 0.5 counts per million. We applied the Wald test to assess differences between experimental conditions. We set *p* ≤ 0.1 after false discovery rate (FDR) adjustment and absolute log2FC > 0.585, annotated the results using the “annotables” package of Bioconductor, and conducted a pathway enrichment analysis against KEGG and Reactome using the “clusterProfiler” package of Bioconductor. The STRING database v12 [27] was used to analyze protein–protein interaction networks of BRD3 target genes overlapping in unstimulated and TNF-stimulated FLS.

### 2.5. Analysis of BRD2, BRD3 and BRD4 Expression in Subsets of FLS

We used publicly available single-cell RNA sequencing (scRNA-seq) data sets available at the BroadSingleCellPortal (https://singlecell.broadinstitute.org/single_cell/study/SCP738, accessed on 19 July 2023) to evaluate BRD2, BRD3, and BRD4 expression in different subtypes of FLS. Inflammation scores were defined as the proportion of immune cells infiltrating each tissue sample [28].

### 2.6. Treatment of FLS

FLS were treated with 1 µM of I-BET151 (Tocris, Bristol, UK), or matched amounts of DMSO, for 24 h. TNF (10 ng/mL) was added simultaneously where indicated. To study BRD3 regulation, FLS were treated with 4-hydroxynonenal (5 µM, 4-HNE; Sigma-Aldrich, Saint Louis, MI, USA) or matched amounts of ethanol for 48 h. To study autophagic flux, FLS were treated with chloroquine (100 µM, CQ; Enzo Life Sciences, Farmingdale, NY, USA) in the absence and presence of I-BET for 16 h, or with bafilomycin A1 (100 nM, Baf; Sigma-Aldrich) for the last 4 h of treatment.

### 2.7. Western Blotting

FLS were lysed in Laemmli buffer (BioRad). Whole-cell lysates were denatured at 95 °C and separated on sodium dodecyl sulfate polyacrylamide gels. Electro blotting was conducted onto nitrocellulose membranes (Whatman, Maidstone, UK), which were afterward blocked for 1 h with 5% non-fat dry milk in TBS-T (20 mM Tris base, 137 mM sodium chloride, 0.1% Tween 20, pH 7.6). Membranes were probed with antibodies against BRD3, p62, α-tubulin (all from abcam), and LC3B (Cell Signaling). Horseradish peroxidase-conjugated goat anti-rabbit or goat anti-mouse antibodies (Jackson ImmunoResearch, West Grove, PA, USA) were used as secondary antibodies. Signals were quantified using the Western Bright ECL Substrate (advansta) or the ECL WesternBlot Detection Reagents (GE Healthcare) and the Fusion FX7 16.01c Software (Vilber, Marne-la-Vallée, France). Signal intensities of target proteins were normalized to the expression of α-tubulin.

### 2.8. Enzyme-Linked Immunosorbent Assay and L-Lactate Assay

VEGF in cell culture supernatants was measured using the DuoSet ELISA human VEGF kit (R&D Systems) following the manufacturer’s protocol. L-lactate in cell culture supernatants was measured using the L-Lactate Assay Kit (Sigma-Aldrich) following the manufacturer’s protocol. Detection was conducted using the GloMax-Multi+ Detection System (Promega, Madison, WI, USA) and the Instinct Software 1.0.0.0 (Promega).

### 2.9. Autophagy Live Cell Imaging

FLS were seeded at 60–80% confluence in 96-well plates and treated as described above. Autophagy was measured using the CYTO-ID Autophagy Detection Kit (Enzo Life Sciences) following the manufacturer’s instructions. The formation of autophagic vacuoles (green dye) and nuclear staining (Hoechst 33342 dye; blue) was monitored on the CX7 CellInsight High-Content Screening (HCS) Platform (Thermo Fisher Scientific, Waltham, MA, USA). The average spot area (mask used “CirSpotAvgAreaCh2”) and the number of spots (mask used “CirSpotCountCh2”) within the cytoplasm of FLS were collected for analysis from nine images per well, per quadruplicated wells for each experimental condition and FLS donor.

### 2.10. Statistical Analysis

Statistical analysis of data sets was carried out by using the GraphPad Prism program 10.0.2 (GraphPad Software, San Diego, CA, USA). N numbers in all experiments represent biological samples from different patients. Differences between experimental groups were analyzed by analysis of variance (ANOVA) followed by Tukey’s multiple comparison test. Data that were not normally distributed were analyzed by the Friedmann test followed by Dunn’s post hoc multiple comparison test. Data are reported as means ± standard deviations. *p* values < 0.05 were considered significant.

## 3. Results

### 3.1. Silencing of BRD3 in FLS

To study the function of BRD3 and identify BRD3 target genes, we performed RNAseq of FLS after silencing of BRD3 in the absence and presence of TNF. BRD3 silencing in FLS decreased *BRD3* mRNA expression by 53.28% and 53.31% in the absence and presence of TNF. This was paralleled by an increase in *BRD2* and *BRD4* mRNA expression by 48.45% and 36.25% in unstimulated FLS, and by 43.32% and 23.46% in TNF-stimulated FLS (Figure 1a). BRD3 silencing was confirmed by Western blotting (Figure 1b, Appendix A). We detected 297 and 416 differentially expressed genes (±fold change > 1.5; FDR ≤ 0.1) that were affected by BRD3 silencing in unstimulated and TNF-stimulated FLS, respectively (Figure 1c–e). In unstimulated FLS, the number of up-regulated genes upon BRD3 silencing slightly exceeded those of down-regulated genes, with 175 versus 122 genes being affected. In TNF-stimulated FLS, 250 genes were down-regulated and 166 genes were up-regulated by silencing of BRD3. In total, 144 differentially expressed genes overlapped between unstimulated and TNF-stimulated FLS (Figure 1e).

### 3.2. Identification of Pathways Regulated by BRD3

To identify how differential gene expression translates to FLS function, we performed a pathway enrichment analysis against KEGG and Reactome databases (Appendix A). This analysis pointed to a fundamental role of BRD3 in the regulation of “cell cycle”, “DNA replication”, and several “stress response pathways”, including “oxidative stress induced senescence”, “DNA repair”, “apoptosis”, and “chaperone mediated autophagy” (Figure 2a–c; Appendix A). Several of these pathways were shared between untreated and TNF-stimulated FLS, as additionally indicated by a STRING database analysis based on the 144 overlapping differentially expressed genes between the two conditions (Appendix A). In this analysis, the majority of proteins were enriched in either “response to unfolded protein”, “cell cycle process”, “structural constituent of chromatin”, or “stress response”.

Although not among the top enriched pathways, BRD3 regulated additionally several inflammatory pathways, including “cytokine signaling in immune system”, “interleukin-7 signaling”, “interferon signaling”, and “Toll-like receptor cascades”.

### 3.3. BRD3 Expression Is Increased in Sublining FLS

The anti-inflammatory effects of BET protein inhibition in RA FLS are well established [16,17,19,29,30]. However, individual BET proteins underlying these effects have hardly been investigated. We first evaluated the expression of different BET proteins in different subtypes of FLS using publicly available scRNA-seq data sets [28]. One type of lining and three types of sublining FLS were defined in synovial tissues (Figure 3a) [28]. The expression of *BRD3*, like *BRD2* and *BRD4*, was increased in sublining FLS compared to lining FLS. *BRD3* expression was equally distributed between different types of sublining FLS (Figure 3b). The mean expression of *BRD3* in FLS, as well as the percentage of *BRD3*-positive cells, moderately correlated with inflammatory scores of synovial tissues (Figure 3c–e). In contrast, neither mean levels of *BRD2* (r = 0.3830; *p* = 0.0934) or *BRD4* (r = 0.2244; *p* = 0.3416) expression, nor percentages of *BRD2*- (r = 0.3755; *p* = 0.1028) or *BRD4*-positive cells (r = 0.2770; *p* = 0.2371) correlated with synovial inflammatory scores (Appendix A).

### 3.4. BRD3 Regulates Inflammatory Pathways in FLS

To analyze the role of BRD3 in regulating inflammatory pathways in FLS in more detail, we selected some of the related genes from our RNAseq data sets and performed real-time PCR experiments. In line with the RNAseq results, silencing of BRD3 suppressed the TNF-induced expression of *CCL2*, *CXCL1*, *CXCL2*, and *ICAM1* (Figure 4). We detected a tendency towards a down-regulation of TNF-induced *IL8* expression. In addition, we observed in unstimulated and TNF-stimulated FLS a tendency towards a BRD3-mediated regulation of *CDK1*, a key cell cycle regulator (Figure 4). Together these data further support our findings from RNAseq and confirm the role of BRD3 in regulating inflammatory gene expression and cell cycle regulation.

### 3.5. BRD3 Regulates Stress-Related Genes in FLS

Since our pathway enrichment analysis pointed to a pronounced role of BRD3 in regulating stress response, we investigated this aspect further. In line with our RNAseq results, silencing of BRD3 induced the expression of *FOS*, and the DnaJ heat shock protein family (Hsp40) family members *DNAJB1* and *DNAJA4*. Furthermore, we observed a potential joint-specific regulation of *CAT* (Figure 5a). Whereas BRD3 silencing suppressed the expression of *CAT* by 50.0% in unstimulated and by 42.3% in TNF-stimulated FLS from hand joints, BRD3 silencing did not affect *CAT* expression in FLS from shoulder joints.

The activation of stress response genes is concomitant with the regulation of metabolic genes in FLS. We detected a BRD3-dependent down-regulation of the TNF-induced mRNA expression of *VEGFA* and a reduced secretion of VEGF into cell culture supernatants in I-BET-treated FLS (Figure 5a,b). Silencing of BRD3 suppressed the expression of *LDHA* in unstimulated and TNF-stimulated FLS and decreased the expression of TNF-induced *PDK1* (Figure 5a). Together, these data suggest that silencing of BRD3 reduced glycolysis. In line with these data, I-BET treatment of FLS suppressed TNF-induced levels of lactate in cell culture supernatants (Figure 5c).

### 3.6. Oxidative Stress Suppresses the Expression of BRD3

To study the regulation of BRD3 upon oxidative stress, we stimulated FLS with 4-HNE, a lipid peroxidation product present in synovial fluids that is known to activate the oxidative stress response [31]. Although not significantly, TNF induced the expression of BRD3 in FLS. In contrast, 4-HNE reduced the TNF-induced expression of BRD3, indicating that BRD3 does not only regulate stress response pathways but is additionally affected by oxidative stress (Figure 6a,b and Appendix A).

### 3.7. BET Proteins Regulate Autophagy in FLS

Given the enrichment of the pathway “Chaperone-mediated autophagy” in our RNAseq data set, including an increased expression of several autophagy-related (ATG) genes and microtubule-associated protein 1 light chain 3 (LC3) genes (Figure 7a), we further evaluated whether BET proteins regulate autophagy in FLS. Since the material of FLS silenced for BRD3 was limited, we stimulated FLS with and without I-BET in the absence and presence of the lysosomal inhibitor bafilomycin or chloroquine (CQ).

I-BET induced levels of LC3B-I to LC3B-II conversion and increased the expression of the autophagy substrate p62 (Figure 7b–d, Appendix A). Bafilomycin further increased levels of LC3B-II and p62, indicating that I-BET induced levels of autophagy. To investigate the effect of I-BET on autophagosome formation and autophagic flux in more detail, we performed live cell imaging in FLS treated with and without I-BET and CQ. In line with the results from our Western blot experiments, I-BET induced autophagosome formation, indicated by an increased fluorescence intensity in the average spot area, in the absence and presence of CQ. In contrast, spot counts were only slightly increased by I-BET, and CQ had similar effects in control and I-BET-treated FLS. Together these data indicate that BET inhibitors induce autophagy in FLS and suggest a role of BRD3 in regulating this process.

## 4. Discussion

BRD3 is an understudied member of the BET protein family, and only a few individual functions have been identified so far. A role of BRD3 in the context of inflammation was suggested in studies with macrophages, in which a BRD3 knockout suppressed the virus-induced IFN-β production [32]. Furthermore, BRD3, together with BRD4, was shown to regulate the expression of matrix metalloproteinases (MMP) under pro-inflammatory conditions in chondrosarcoma cells [33]. Other studies have underscored the role of BRD3 in cell differentiation processes, including endoderm differentiation [34], myogenesis [35], erythroid maturation [36], and embryonic stem cell differentiation [37]. Our transcriptome analysis provides new and so far unknown insights into BRD3 function.

We and others previously identified a key role of BET proteins in regulating FLS proliferation and cytokine, chemokine, and MMP expression [16,17,29,30,38,39]. These studies are largely based on the use of pan-inhibitors, such as I-BET and JQ-1, and individual roles of BET proteins have only been studied for BRD2 and BRD4. Their function has been based on the measurement of selected target genes, including *IL6*, *IL8*, *MMP1*, *MMP3*, and *MMP13*, after silencing of BRD2 and BRD4 in FLS prior to TNF stimulation [16]. Our study adds BRD3 to the regulatory proteins underlying BET inhibitor effects in FLS. We provide the first transcriptome analysis after silencing a member of the BET protein family in FLS. The top pathway regulated by BRD3 was the “cell cycle”, in line with the previously reported effect of BET inhibitors suppressing FLS proliferation [17,38]. The role of BRD3 in regulating proliferation was further underscored by the reduced expression of *CDK1* upon BRD3 silencing.

Here, we provide evidence that at least some of the anti-inflammatory effects of BET inhibitors can be attributed to the function of BRD3. We were able to identify several BRD3-regulated cytokines and chemokines, including *IL8*, that were previously reported to also be regulated by BRD2 and BRD4 [16]. JQ-1 treatment of IL-1-treated FLS has been shown to dislodge BRD2 and BRD4 from chromatin regions, in particular from super-enhancers regulating cytokine expression. Of note, only a small portion of JQ-1-mediated changes in BRD2- and BRD4-occupied chromatin regions overlapped, indicating that JQ-1 action was mediated by distinct BET-regulated chromatin regions [17]. However, the study did not include an analysis of potential JQ-1-mediated effects on BRD3 chromatin occupancy. Such effects and overlaps with BRD2- and BRD4-regulated enhancers are likely, since among the analyzed BRD2- and BRD4-dependent super-enhancers were those for *IL8*, *CCL2*, and *VEGFA*, which we show here to be additional target genes of BRD3.

JQ-1 was shown to reduce the chromatin accessibility in six times as many regions in IL-1-treated FLS compared to untreated FLS, potentially affecting the binding of key transcription factors involved in RA, such as members of the NF-κB and AP-1 families [17]. Therefore, it was surprising that we identified only 1.4 times more target genes of BRD3 in TNF-stimulated compared to unstimulated FLS, and almost half of the target genes identified in unstimulated FLS were additionally found to be BRD3 targets in TNF-stimulated FLS. These numbers suggest a key role of BRD3 in regulatory processes beyond inflammation, which is further underpinned by the fact that inflammatory pathways were not among the top enriched pathways in our RNAseq data sets. Nevertheless, the expression of BRD3 in FLS positively correlated with inflammatory scores in synovial tissues.

Chronic inflammation in the RA joint is accompanied by several metabolic changes, the formation of lactate and increased levels of reactive oxygen species (ROS), leading to oxidative stress [9,40]. We identified BRD3 as a regulator of key steps associated with the metabolic adaption in FLS by regulating the TNF-induced expression of *LDHA*, *PDK1*, and *VEGFA*. Synovial ROS provoke the oxidation of polyunsaturated fatty acids, generating an array of lipid peroxidation products, including 4-HNE. Elevated levels of 4-HNE were found in serum, synovial fluids, and synovial tissues of patients with RA [31,41], and serum levels correlated with structural damage such as erosions in the early stage of RA [41]. We showed here that TNF and oxidative stress, mimicked by the treatment of FLS with 4-HNE, regulate the expression of BRD3. Given that 4-HNE suppressed the TNF-induced expression of BRD3, we suggest that BRD3 is part of a feedback loop that allows the cell to cope with stress conditions in inflammatory conditions.

Among the top BRD3-regulated pathways in unstimulated and TNF-stimulated FLS were different stress response pathways. Adaption to hypoxia, cell survival/death programs, autophagy, and UPR and ER stress response pathways were shown to be activated in the RA synovium [4,5,6,7,8,42]. Furthermore, the activation of autophagy in FLS was shown to enhance levels of citrullinated vimentin, a major autoantigen in RA, and its interaction with MHC class II molecules [43]. BET protein-dependent stress response in FLS has not been studied yet.

We focused on the role of BRD3 in regulating oxidative stress response and autophagy. JQ-1 and silencing of BRD2 and BRD4 were previously described to increase the expression of antioxidant response genes in THP-1 monocytes [44]. Furthermore, JQ-1 and silencing of BRD4 reduced the production of ROS and increased the expression of antioxidant genes, including *CAT*, in H_2_O_2_-challenged rat chondrocytes [45]. In FLS, silencing of BRD3 induced the expression of *CAT* only in FLS from shoulder but not hand joints. We have previously shown joint-specific expression patterns in FLS, and *CAT* expression was increased in FLS from knee joints compared to shoulder and hand joints [23]. To confirm a potential joint-specific regulation of *CAT* expression, samples from more donors would be needed.

We have detected several heat shock proteins to be enriched in stress-related pathways in FLS after the silencing of BRD3 and confirmed *DNAJB1* and *DNAJA4* as BRD3 target genes. *DNAJA4* was previously shown to be induced by stimulation of FLS with cigarette smoke extract [46], a condition that mimics smoking as the most common environmental risk factor in RA [47], and increases oxidative stress in fibroblasts [48]. Among BET proteins, only BRD4 has been described to regulate heat shock response [19].

Oxidative stress is a known inducer of autophagy [49]. We have recently shown that CBP- and p300-mediated protein acetylation regulates oxidative stress response and autophagy at transcriptional and functional levels in FLS [50]. Here, we add BRD3, as a reader of acetylated side chains, as an additional regulator of both processes. Whereas BRD4 has been described as a repressor of autophagy [21], the role of BRD3 in this process was so far unknown. Similar to results from silencing of BRD4 in KP-4 cells [21], we identified several autophagy-related genes to be induced upon silencing of BRD3 in FLS. In line with these findings, we showed that I-BET induced levels of autophagy in FLS. The use of I-BET in functional autophagy experiments is one limitation of this study. We cannot rule out that other members of the BET protein family are additionally involved in the regulation of autophagy in FLS.

Since we have not included other BET proteins in this study, we cannot compare the effects of BRD3 silencing to the individual effects of other BET proteins. We showed that the silencing of BRD3 led to a compensatory up-regulation of *BRD2* and *BRD4*, at least on mRNA levels. Therefore, some of the observed effects might be related to other BET proteins. Among the BRD3 target genes identified were several structural constituents of the chromatin. Previous studies illustrated by ATAC-sequencing and mass spectrometry that BET inhibitors remodeled the chromatin of FLS [17,30,39]. Since we have not studied potential effects on chromatin structure, we cannot rule out that the silencing of BRD3 induced changes in promoter and enhancer accessibilities of identified target genes.

## 5. Conclusions

BRD3 acts as an upstream regulatory factor of the cell cycle and of inflammatory and stress pathways in FLS. Thus, BRD3 executes many functions of BET proteins that have previously been identified using pan-BET inhibitors.

## Figures and Tables

**Figure 1 biomedicines-11-03188-f001:**
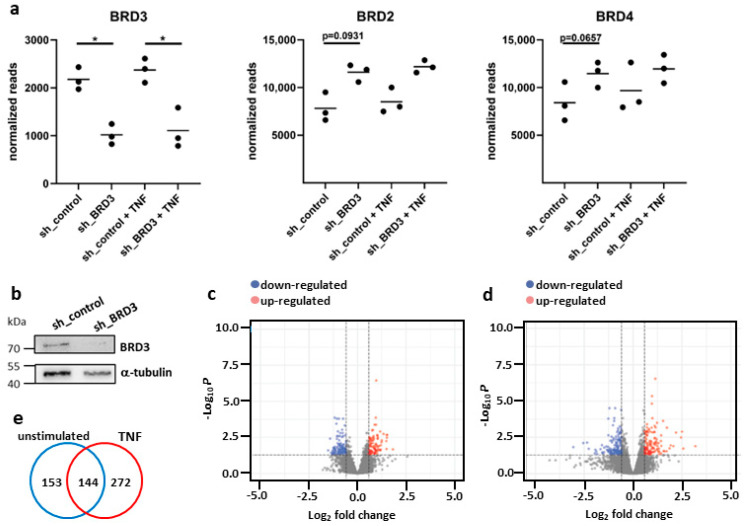
Silencing of BRD3 in FLS. The expression of BRD3 was silenced in FLS from wrist joints by transduction of lentiviral particles prior to stimulation with TNF (10 ng/mL). Transcriptomes were analyzed by RNAseq. (**a**) Expression levels of *BRD3*, *BRD2*, and *BRD4* after silencing of BRD3 in RNAseq data sets. (**b**) Silencing of BRD3 was confirmed by Western blotting. Volcano plots of RNAseq data sets in (**c**) unstimulated and (**d**) TNF-stimulated FLS after silencing of BRD3. (**e**) Venn diagram of BRD3-regulated genes (±fold change > 1.5; FDR ≤ 0.1) in unstimulated and TNF-stimulated FLS. * *p* < 0.05.

**Figure 2 biomedicines-11-03188-f002:**
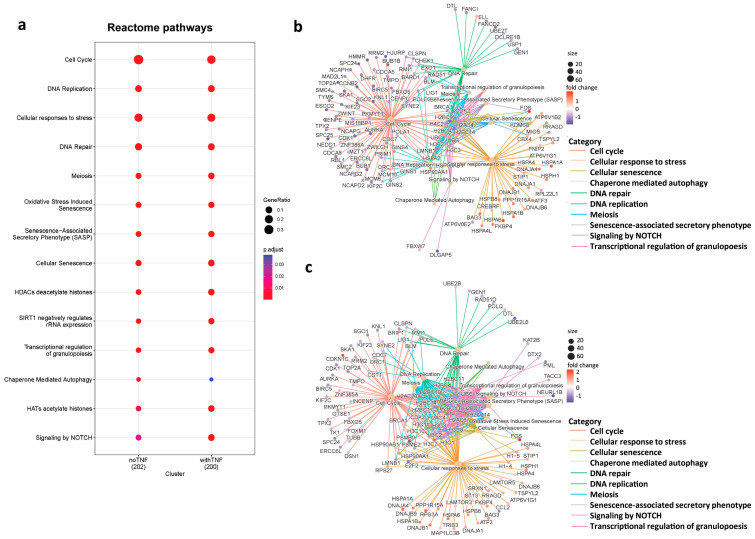
Pathway enrichment analysis of FLS silenced for BRD3. Differentially expressed genes (±fold change > 1.5; FDR ≤ 0.1) of FLS silenced for BRD3 entered pathway enrichment analysis. (**a**) Reactome pathways shared between unstimulated and TNF-stimulated FLS are shown. The full list of enriched pathways, and genes enriched in these pathways, can be found in Appendix A. cnet plots depicting the linkages of genes in enriched pathways (**b**) in unstimulated and (**c**) TNF-stimulated FLS.

**Figure 3 biomedicines-11-03188-f003:**
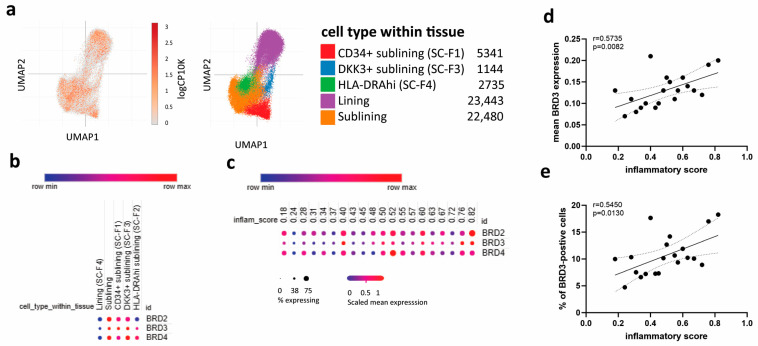
Expression of BET proteins in subtypes of FLS. (**a**) Visualization of different subtypes of FLS by UMAP using scRNA-seq data sets available at the BroadSingleCellPortal [28]. (**b**) *BRD2*, *BRD3*, and *BRD4* expression in subtypes of FLS. (**c**) Dot plots indicating *BRD2*, *BRD3*, and *BRD4* expression in synovial tissues with different inflammatory scores as defined by inflammatory cell infiltration. Correlation of (**d**) mean *BRD3* expression and (**e**) percentage of *BRD3*-positive cells with inflammatory scores of synovial tissues.

**Figure 4 biomedicines-11-03188-f004:**
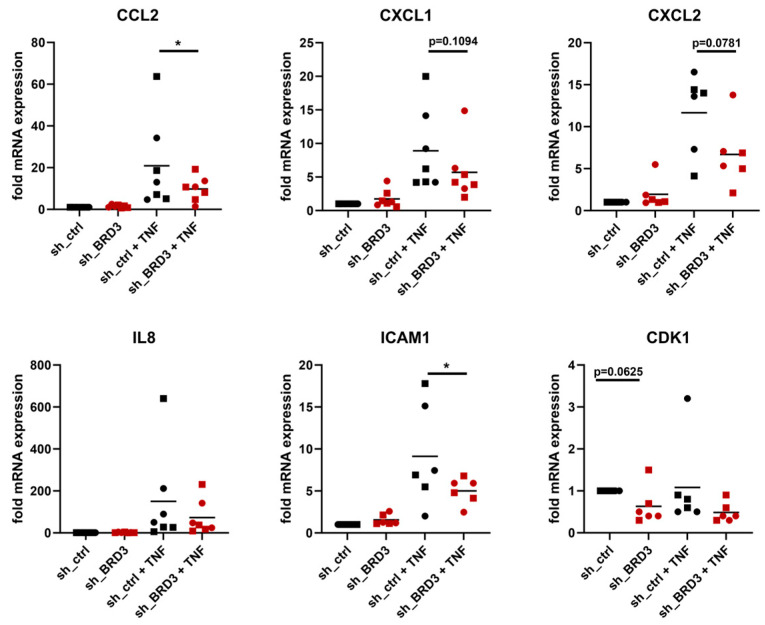
Inflammatory target genes of BRD3. FLS from hand (circles) and shoulder (squares) were silenced for BRD3 and stimulated with TNF or left untreated. The expression of inflammatory genes, identified to be differentially expressed by RNAseq, was measured by real-time PCR. * *p* < 0.05.

**Figure 5 biomedicines-11-03188-f005:**
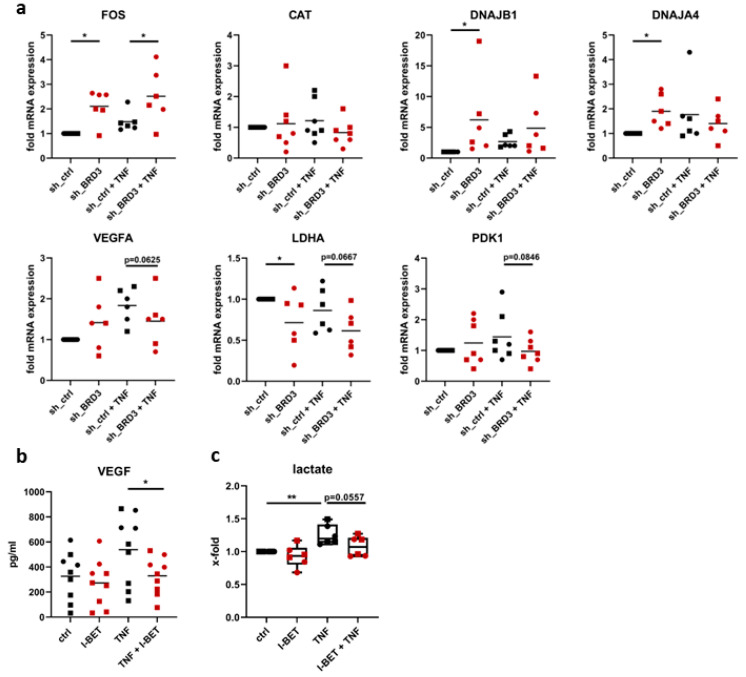
BRD3-regulated stress response in FLS. (**a**) FLS from hand (circles) and shoulder (squares) were silenced for BRD3 and stimulated with TNF or left untreated. The expression of stress response-related genes was measured by real-time PCR. FLS from hand (circles) and shoulder (squares) were treated with I-BET in the absence and presence of TNF. (**b**) VEGF secretion and (**c**) extracellular lactate levels in cell culture supernatants. * *p* < 0.05; ** *p* < 0.01.

**Figure 6 biomedicines-11-03188-f006:**
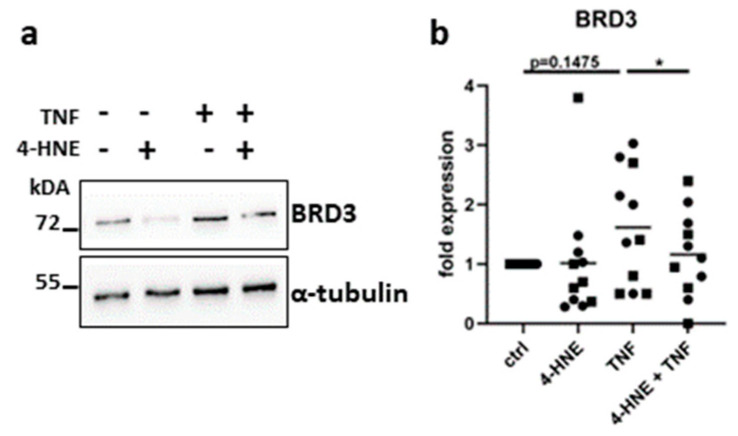
Oxidative stress regulates the expression of BRD3 in FLS. FLS from hand (circles) and shoulder (squares) were stimulated with 4-HNE in the absence and presence of TNF. The expression of BRD3 was measured by Western blotting. (**a**) A representative Western blot is shown. (**b**) Densitometric analysis of Western blot results. * *p* < 0.05.

**Figure 7 biomedicines-11-03188-f007:**
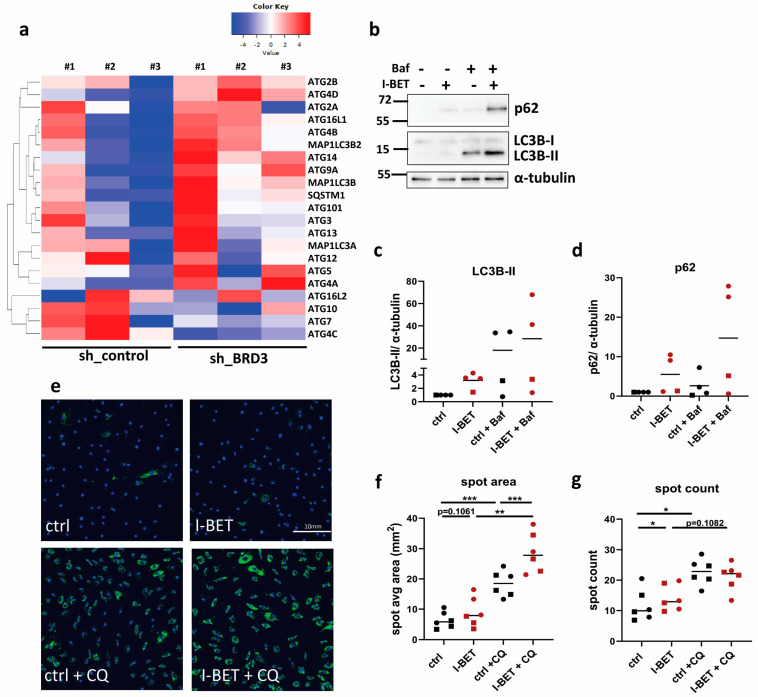
I-BET induces autophagy in FLS. (**a**) Heatmap of autophagy-related genes identified by RNAseq in FLS silenced for BRD3. (**b**) FLS from hand (circles) and shoulder (squares) joints were treated with I-BET in the absence and presence of bafilomycin (Baf). The conversion of LC3B-I to LC3B-II and the expression of p62 and α-tubulin were measured by Western blotting. A representative Western blot is shown. Densitometric analysis of Western blot results for (**c**) levels of LC3B-II and (**d**) p62. (**e**) Formation of autophagosomes in I-BET-treated FLS (*n* = 6) was measured by live cell imaging in the absence and presence of chloroquine (CQ). Representative images for each condition (in shoulder FLS) are shown. (**f**) Average spot areas and (**g**) average spot counts were calculated from quadruplicates for each condition and patient sample. * *p* < 0.05; ** *p* < 0.01; *** *p* < 0.005.

## Data Availability

Publicly available scRNA-seq data sets were analyzed using the BroadSingleCellPortal [28]. The RNAseq data of FLS silenced for BRD3 in the absence and presence of TNF generated in this study have been deposited in NCBI’s Gene Expression Omnibus [24] and are accessible through the GEO Series accession number GSE247226 (https://www.ncbi.nlm.nih.gov/geo/query/acc.cgi?acc=GSE247226). Other data sets generated and/or analyzed during the current study are available from the corresponding author upon reasonable request.

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
