# Peer review of "BRD3 Regulates the Inflammatory and Stress Response in Rheumatoid Arthritis Synovial Fibroblasts"

_biomedicines, 2023, doi:10.3390/biomedicines11123188_

Round 1

Reviewer 1 Report

Comments and Suggestions for Authors

In the reviewed study, the authors have set a clear goal. The methodology of the work is not questionable. In the introduction, the aim of the study was warranted. I have no objections to the methodology and ethical aspects. The only comment relates to limitations. I believe that this paragraph should be expanded to include broader aspects of factors that could affect the results obtained. This part of the paper is extremely important and should be expanded.

Author Response

Our response: We have added a paragraph at the end of the discussion section to mention further 
limitations of the study. This includes comments on the compensatory upregulation of BRD2 and 
BRD4, and potential effects on chromatin structure upon BRD3 silencing. Since we do not know what 
exactly the reviewer had in his/ her mind, we hope that we have sufficiently addressed this point.

Reviewer 2 Report

Comments and Suggestions for Authors

In the manuscript titled “BRD3 regulates the inflammatory and stress response in rheumatoid arthritis synovial fibroblasts”, Tanja Seifrit et al., performed Real-time PCR, ELISA and Western blotting and demonstrated regulatory role of BRD3 in RA synovial fibroblasts. This study contains some interesting findings that are valuable for the understanding of function of BRD3 in RA. However, the imperfect format and incomplete data are the main defects of this study. Therefore, minor revision has to be done before this manuscript could be accepted for publication in the Biomedicines.

Major comments:

1.       Line 205: “In contrast, …… (data not shown)”, please provide references or data to prove it.

2.       In the “Conclusion” part, drawing too broad a conclusion about the work done and too brief a description of the BRD3 protein studied does not show the importance of the research.

Minor comments:

1.       Line 35-37, one recent review (doi.org/10.1002/EXP.20220132) should be included to support such a claim.

2.       The resolution of figures 2-3 should be improved to a much higher level.

3.       The text and content in Supplementary Figure 1 are not clear.

4.       Supplementary Figure 23 and 4, the captions are too small.

Comments on the Quality of English Language

Minor editing of English language required

Author Response

Major comments:
1. Line 205: “In contrast, …… (data not shown)”, please provide references or data to prove it.
Our response: We have provided a new supplementary figure (S3) were we provide the data. We have relabelled the other supplementary figures accordingly.
2. In the “Conclusion” part, drawing too broad a conclusion about the work done and too brief a description of the BRD3 protein studied does not show the importance of the research.
Our response: We have rephrased the conclusion to “BRD3 acts as an upstream regulatory factor of the cell cycle, inflammatory and stress pathways in FLS. Thus, BRD3 executes many functions of BET proteins that have previously identified using pan-BET inhibitors.” We believe that we do not make the story more important than it is and we do not claim in the conclusion anything that is 
not shown. With the brief conclusion section, we follow the suggested style by the journal. 
Minor comments:
1. Line 35-37, one recent review (doi.org/10.1002/EXP.20220132) should be included to suport such a claim.
Our response: We have cited the suggested review (new reference 3). Additionally, we have clarified that fibroblasts are the main cellular cell type in lining and sublining layers in the RA
synovium.

2. The resolution of figures 2-3 should be improved to a much higher level.
Our response: We have improved the resolution of figures 2 and 3. 

3. The text and content in Supplementary Figure 1 are not clear.
Our response: We have extended the respective figure legend (now Figure S2) and have added another sentence in the text of the manuscript. We hope that it is now clear that the figure provides an additional pathway analysis using only the overlapping BRD3 target genes in unstimulated and TNF-stimulated FLS (as identified in Fig. 1e).

Reviewer 3 Report

Comments and Suggestions for Authors

Dear authors

The topic of this is a good. The authors have assessed the role of BRD3 in rheumatoid arthritis FLS using various valid tests.

In addition, the writing of the manuscript is acceptable.

The journal style for the manuscript preparation can be considered.

Mechanisms of BRD3 has been demonstrated which are acceptable.

The discussion section is appropriate.

The references are updated.

Also follow the journal style for the manuscript preparation.

Best regards

Author Response

Many thanks for your comments. 
